# Improvement of the Photocatalytic Activity of Au/TiO$_2$ Nanocomposites by Prior Treatment of TiO$_2$ with Microplasma in an NH$_3$ and H$_2$O$_2$ Solution

**Nguyen Thi Thu Thuy [1,2], Do Hoang Tung [3,\*], Le Hong Manh [3], Pham Hong Minh [3] and Nguyen The Hien [1]**

[1] Faculty of Engineering Physics and Nanotechnology, VNU University of Engineering and Technology, 144 Xuan Thuy Road, Cau Giay District, Hanoi 10000, Vietnam; thuthuyt34@gmail.com (N.T.T.T.); thehien@vnu.edu.vn (N.T.H.)

[2] Faculty of Basic Science, University of Fire Prevention and Fighting, 243 Khuat Duy Tien, Thanh Xuan, Hanoi 10000, Vietnam

[3] Institute of Physics, Vietnam Academy of Science and Technology, 18 Hoang Quoc Viet Street, Cau Giay District, Hanoi 10000, Vietnam; levietvuk6t34@gmail.com (L.H.M.); phminh@iop.vast.ac.vn (P.H.M.)

\* Correspondence: trunghoct34@gmail.com

**Abstract:** Plasmonic photocatalytic nanocomposites of TiO$_2$ and Au nanoparticles (NPs) have recently attracted the attention of researchers, who aim to improve the photocatalytic activity of potential TiO$_2$ NPs. In this study, we report photocatalytic activity enhancement for a Au/TiO$_2$ nanocomposite prepared by the plasma–liquid interaction method using an atmospheric microplasma apparatus. The enhanced photocatalytic activity of the prepared Au/TiO$_2$ is demonstrated by the degradation of methylene blue (MB) in water under both ultraviolet (UV) and visible light irradiation. The prior treatment of TiO$_2$ with microplasma in a NH$_3$ and H$_2$O$_2$ solution is found to strongly improve the photocatalytic activity of both the treated TiO$_2$ NPs, as well as the synthesized Au/TiO$_2$ nanocomposite.

**Keywords:** nanocomposite; microplasma–liquid; photocatalytic

## 1. Introduction

Photocatalysts can effectively decompose various organic pollutants via catalysis using photons. In recent years, they have attracted great interest as potential candidates for effective applications in solar water treatment systems, with advantage such as being greatly abundant, easy and cheap to fabricate, and non-toxic, in addition to possessing high chemical stability and high optical transparency, TiO$_2$ NPs have shown up as a promising efficient photocatalyst material [1,2]; however, the biggest disadvantage of TiO$_2$ NPs in this respect is that their large band energy only allows materials to be effective photocatalysts if they are within an ultraviolet (UV) region. Therefore, the photocatalysis of TiO$_2$ NPs cannot be realized with normal sunlight [3].

It is well understood in the photocatalytic degradation of TiO$_2$ NPs, that on the one hand, the dye molecules adsorbed are directly oxidized by the photo-generated holes on the surface of TiO$_2$ NPs, or in another way, i.e., indirect oxidation by photocatalyzed generated hydroxyl (OH$^\bullet$) radicals. This process is described as follows: In water or in an aqueous solution, the absorption of UV photons can generate electron–hole pairs in TiO$_2$ NPs:

$$\text{TiO}_2 + h\nu \ (\text{UV}) \rightarrow \text{TiO}_2 * (e^- + h^+) \tag{1}$$

The hydroxyl radicals can be generated through either the oxidation pathway of water by the hole,

$$\text{H}_2\text{O} + h^+ \rightarrow \text{OH}^\bullet + \text{H}^+ \tag{2}$$

$$\text{OH}^-_{\text{ads}} + h^+ \rightarrow \text{OH}^\bullet, \tag{3}$$

or through the reduction pathway of oxygen by the electron [4,5],

$$O_2 + e^- \rightarrow O_2^{\bullet-} \tag{4}$$

$$O_2^{\bullet-} + H^+ \leftrightarrow HO_2^{\bullet} \ (pKa = 4.88) \tag{5}$$

$$2\,HO_2^{\bullet} \rightarrow H_2O_2 + O_2 \tag{6}$$

$$H_2O_2 + h\nu \ (UV) \rightarrow 2OH^{\bullet} \tag{7}$$

The superoxide ions reduced from the oxygen by the electron (Equation (4)) can change to $HO_2^{\bullet}$, $H_2O_2$, and then to hydroxyl radicals in sequence (Equations (5)–(7)). Then, these hydroxyl ($OH^{\bullet}$) radicals disperse near the $TiO_2$ NP surface and promote the oxidation of MB molecules. Due to their large bandgap energy, the $TiO_2$ NPs do not absorb visible light that is able to support the aforementioned processes.

Different approaches to widening the absorption spectrum of $TiO_2$ towards visible wavelengths, including hydrogenation or metal/nonmetal doping, have been reported [6–9]; however, these efforts have a disadvantage in creating numerous defects, such as causing high electron–hole recombination rates, which harmfully affect the photocatalytic function [10].

Nanocomposites of noble metals and $TiO_2$ (NPs) have been considered as potential materials for the improvement of photocatalytic efficiency in this regard [10–13]. Localized surface plasmon resonance (LSPR) studies have revealed a remarkable blue shift in the case of noble metal nanoparticles and it has been shown that these nanoparticles can strongly absorb visible light and inject hot electrons into a $TiO_2$ matrix to induce photocatalytic activity under visible light [12,14,15]. In these cases, the hot electrons generated from the non-radiative decay of the surface plasmon of the AuNPs can be injected into the conduction band of the $TiO_2$ NPs. The injected electrons in the $TiO_2$ NPs and the remaining holes in the AuNPs then act in the reduction and oxidation processes, respectively, to form the hydroxyl radicals [13]. With the electrons and holes created this way, the obstacle of the large bandgap of $TiO_2$ in the catalytic reaction may be overcome. Furthermore, AuNPs also reduce electron-hole recombination rates by way of charge separation, and, thus, can further enhance the photocatalytic activity [12].

The attachment of AuNPs onto the surface of the $TiO_2$ NPs has been attempted with various techniques, such as physical adsorption, photo-deposition, and deposition precipitation [16–22]. Compared to these techniques, however, plasma-liquid interaction is a very simple, fast, and eco-friendly method that can reduce AuNPs from precursors directly onto the surface of $TiO_2$ NPs. Indeed, in plasma–liquid interactions, atmospheric plasma and microplasma can produce many energetic electrons that are added into the solution for the reduction of $Au^{3+}$ ions without the need for chemical-reducing agents [23–26]. Microplasma has been successfully deployed for the synthesis of AuNPs, both in solution and/or directly attached to the surface of dispersed dielectric NPs in solution, as reported in previous works [24,26–28]. The profound advantage of this method is that bare AuNPs can be directly attached to the surface of $TiO_2$ NPs without any chemicals present between them. This also facilitates electron transport and thus enhances the photocatalytic efficiency.

The generation of AuNPs on the surface of $TiO_2$ NPs by plasma–liquid interaction is governed by the so-called microplasma-induced liquid electrochemistry, involving the following three pathways of the reduction of AuNPs from $HAuCl_4$.

Firstly, plasma-generated electrons, solvated in water directly, reduce the Au cations, where Au seeds are then nucleated and grow into AuNPs [27–29]. The process is expressed as follows:

$$Au^{3+} + 3e^- \rightarrow Au^0 \tag{8}$$

Secondly, the interaction between the plasma electrons and water molecules at the plasma-liquid interface generates hydroxyl ($OH\bullet$) radicals and a subsequent reaction in

solution where $H_2O_2$ is rapidly formed. The $H_2O_2$ can then reduce the Au ions from the precursor [30–33] via the following reactions:

$$e^-_{\text{plasma}} + H_2O \rightarrow OH\bullet + H^- \tag{9}$$

$$2OH\bullet \rightarrow H_2O_2 \tag{10}$$

$$3H_2O_2 + 3OH^- + Au^{3+} \rightarrow Au^0 + 3HO_2 + 3H_2O \tag{11}$$

Lastly, the plasma-generated UV photons are absorbed by the $TiO_2$ NPs and this induces electron-hole pairs on the surface of the $TiO_2$ NPs. These electrons and holes can also interact in the direct reduction of Au ions on the $TiO_2$ surface, or to generate radicals that can subsequently interact to reduce the Au ions [29,34,35].

In the present work, $TiO_2$ NPs are functionalized and a $Au/TiO_2$ nanocomposite is synthesized by means of atmospheric microplasma-liquid interaction. The photocatalytic activity of the as-prepared nanocomposites has been studied via the degradation of methylene blue (MB) dye under UV and visible light irradiation. The results of the experiments show the enhanced catalytic performance of the prepared NPs when compared with that of the pristine $TiO_2$ NPs, under both UV and visible light irradiation. In order to understand the mechanism of the photocatalytic degradation of MB dyes by the $Au/TiO_2$ nanocomposite, the degradation of MB dyes under UV and visible light irradiation is also systematically studied here by changing the pH value of the solution.

## 2. Experimental Methods

For the synthesis of the $Au/TiO_2$ composites, commercial sub-micron (averaged size ca. 800 nm) rutile $TiO_2$ particles (R800) bought from US Research Nanomaterials, Inc., Houston, TX 77084, USA, Au(III) chloride trihydrate (99.9%; $HAuCl_4 3H_2O$) purchased from Sigma-Aldrich, and a 30% hydrogen peroxide solution and 25%–28% ammonia solution purchased from Xilong Scientific Co., Ltd., Chaoshan Road, Shantou, Guangdong 510663, China, were used without any further treatments.

The microplasma–liquid reaction system was composed of a plasma nozzle, a gold electrode, and a Bomex glass beaker of a 50 mL volume. The plasma was operated with a DC high-voltage power supply. In the experiments, treatment solutions of 20 mL were used and were contained in a Bomex glass beaker (see Figure 1). The plasma nozzle was constructed with a Teflon tube, of which one end was a hollow cylinder of 15 mm for the outer diameter and 10 mm for the inner diameter, housing the plasma electrode, and the other end was connected to a 6 mm tube for quick gas supply. The arc discharge electrode was a 1.6 mm diameter tungsten rod with a sharpened tip. The rod was connected to one of the poles of the high voltage supply unit via a 100 k$\Omega$ resistor. The other pole of the high voltage source was grounded together with a gold electrode which was submerged into the treatment solution. The processing plasma current was maintained constantly at 5 mA.

The treated $TiO_2$ NP sample was prepared by the functionalization of the $TiO_2$ NP surface by exposing the 20 mL solution of $TiO_2$ 0.1 g·L$^{-1}$, 3% $NH_3$, and 3% $H_2O_2$ to a positive plasma (Figure 1b) source for 15 min. The resulting solutions were subsequently filtered and washed with DI water several times, then vacuum dried to eliminate all remaining $H_2O_2$ and $NH_3$. Finally, DI water was added to obtain a 20 mL solution of treated $TiO_2$ with a 0.1 g·L$^{-1}$ concentration.

To synthesize the $Au/TiO_2$ nanocomposite, 20 mL of the solution of $TiO_2$ 0.1 g·L$^{-1}$ and $HAuCl_4$ 0.06 mM was created then reacted with negative plasma (configuration shown in Figure 1a). It usually takes 15 min for the reduction of the Au(III) in the solution to be completed and to obtain the desired Au/treated $TiO_2$ nanocomposites.

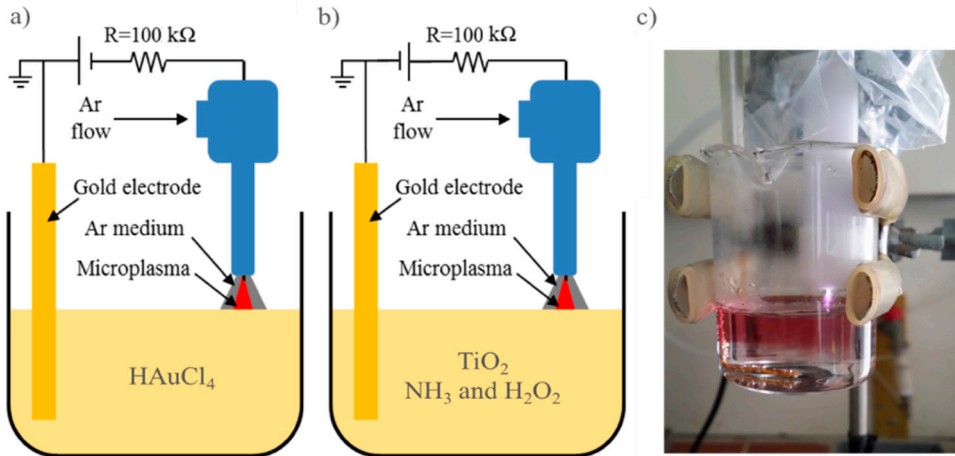

**Figure 1.** Micro-plasma discharge workstation for (**a**) the electrochemical gold reduction, (**b**) the TiO$_2$ treatment, and (**c**) the image of the running plasma for gold nanoparticle synthesis from a HAuCl$_4$ 0.06 mM precursor solution.

The surface morphology of the Au/TiO$_2$ nanocomposite sample was examined by a field emission scanning electron microscope (using the FE-SEM, Hitachi, Tokyo, Japan, S4800). The UV-visible absorption spectrum was measured using an UV-near IR spectrometer (Jasco V-570). The zeta potential values of the NPs were measured using the Zetasizer Nano ZSP from Malvern Instruments.

The photocatalytic activity of the treated TiO$_2$ NPs and the Au/TiO$_2$ nanocomposite in the MB degradation was studied here.

Firstly, for the experiment with the treated TiO$_2$ NPs, the sample was prepared in a 100 cm$^3$ cuvette by mixing three milliliters (3 mL) of the prepared photocatalysts and MB solution with the final NPs and MB with concentrations of 0.1 g/L and 20 µM, respectively. The photocatalytic reaction for the MB degradation in the cuvette was induced by a collimated LED light with a specific wavelength range that was at incident from above the solution. The solution was continuously stirred with a magnetic stirrer. The UV light was provided by a UV LED (Thorlabs, Newton, NJ, USA, M365L2-C1) which emitted UV light in a wavelength range from 354 to 385 nm with peak intensity at 365 nm. The visible light was produced by a Thorlabs M530L3-C1 LED. This LED emitted green light in the wavelength range from 484 to 562 nm with peak intensity at 513 nm. The light power from the LED was measured at the focusing point on the surface of the solution in the measuring cuvette to be 20 mW for both LEDs.

Figure 2 presents the UV-vis absorption spectra of the MB aqueous solution. The UV absorption spectrum (a) shows a double-peak feature at 664 and 613 nm, corresponding to monomers and dimers, respectively [35,36]. For the evaluation of the photocatalytic activity of the TiO$_2$ NPs, the variation of the intensity of the characteristic peak at 664 nm was recorded. Because the UV absorbance peak intensity is directly proportional to the MB concentration in the sample solution, the change of the MB concentration during a photocatalytic degradation reaction is clearly evident as per the decrease of the intensity of this characteristic absorbance peak at 664 nm as the degradation of MB preceded. In the case of the visible light irradiation (b), the 664 nm peak intensity of the absorbance spectrum of the TiO$_2$ NPs-MB solution indicated almost no change.

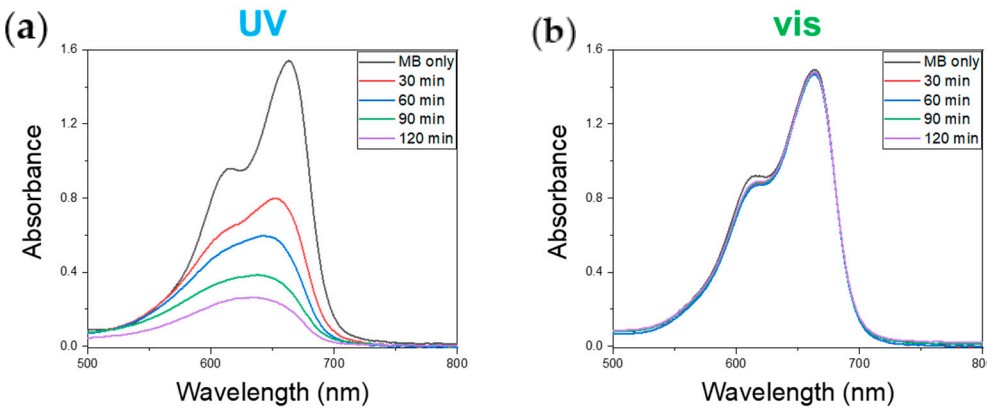

**Figure 2.** Photocatalytic degradation of MB with $TiO_2$ NPs under: (**a**) UV and (**b**) visible irradiation.

In this work, the photocatalytic degradation of MB dye was furthermore investigated at various conditions. Firstly, the solution pH values of 3, 6.5 and 10, corresponding to acidic, neutral, and basic conditions, respectively, were considered. The pH values were monitored by a pH meter (portable multi-item water quality meter—HI98494) while adding sodium hydroxide and hydrochloric acid to the solution. The MB degradation with both the treated $TiO_2$ NPs and $Au/TiO_2$ nanocomposites was then also examined with different irradiation light wavelengths.

## 3. Results and Discussion

Figure 3 depicts the scanning electron microscope (SEM) images of the synthesized $Au/TiO_2$ nanocomposite. As can be clearly seen, the AuNPs were intimately attached to the surface of the $TiO_2$ NPs without obvious aggregation.

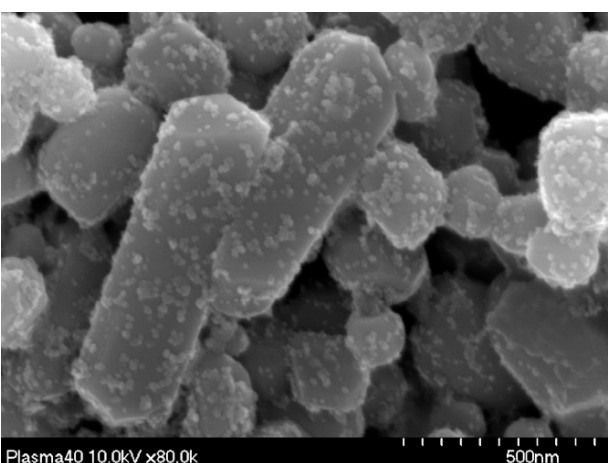

**Figure 3.** SEM images of synthesized Au/treated $TiO_2$ nanocomposite.

It is well known that the electrostatic properties of NPs can cause dramatic changes in their photocatalytic efficiency. This is because of the electrostatic interaction on the NP surface between NPs and the charged molecules in their proximity. The electrostatic property of pristine $TiO_2$ NPs, the plasma treated $TiO_2$ NPs, and the synthesized $Au/TiO_2$ nanocomposite were investigated by measuring their zeta potentials at different pH values and the results are shown in Figure 4.

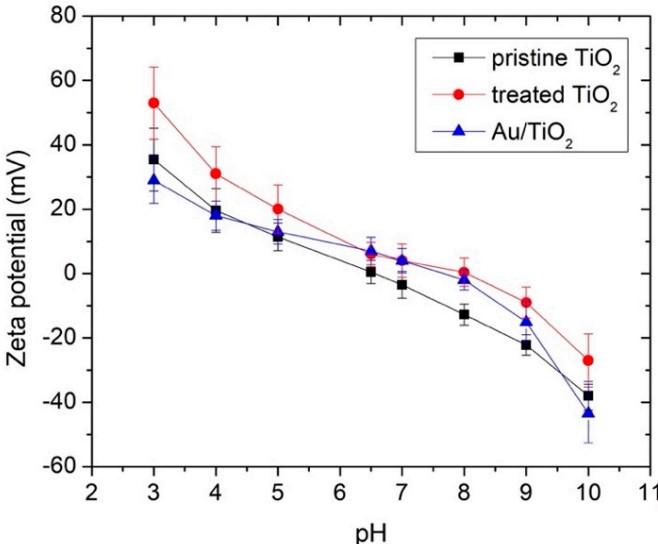

**Figure 4.** Measured zeta potential values of pristine $TiO_2$, plasma-treated $TiO_2$, and the $Au/TiO_2$ nanocomposite at various pH values.

As can be seen, the zeta potential curve obtained for the treated $TiO_2$ NPs exhibits almost the same behavior as that of the pristine $TiO_2$ NPs, but levelled up by about 8–19 mV. This suggests that there could be a certain amount of positive nitrogenic groups positioned on the $TiO_2$ NP surface. The measured curve for the $Au/TiO_2$ nanocomposite clearly shows changes in the zeta potential with respect to the pH values when compared to those of the other two samples. Upon the implantation of the AuNPs on the treated $TiO_2$ NPs, whereas the AuNPs were more likely to be attached to nitrogenic groups, these seem to minimize their contribution to the surface electrostatic potential of the composite.

The well-pronounced correlation between the zeta potential of the nanomaterials and the pH values may also be seen in Figure 4. When the pH value was increased from 3, i.e., from an acidic condition, to 10, i.e., a highly basic condition, the zeta potential of the plasma-treated $TiO_2$ changed from highly positive (about +(53–54) mV) to moderately negative (about −(28–29) mV) while that of the $Au/TiO_2$ nanocomposite changed from moderately positive (about +(29–30) mV) to highly negative (about −(46–47) mV). This means that the electrostatic interaction between the plasma-treated $TiO_2$ and the cationic MB molecules has changed from a highly repulsion situation to slight attraction. In accordance with this, the local MB concentration close to the NP surface would be small at pH = 3, whereas it would be much higher at pH = 10. The very small zeta potentials at nearly neutral pH values from 6.5 to 7 would denote that the NPs neither repulse nor attract the MB molecules.

The photocatalytic efficiency of the pristine $TiO_2$ NPs, the plasma-treated $TiO_2$ NPs, and the as-synthesized $Au/TiO_2$ nanocomposite was examined. The results are presented in Figure 5 in terms of the relative concentration ($C/C_0$ in %) versus measuring time (in minutes) for comparison and evaluation. Measurements of the MB concentration of the sample solution were carried out while measuring the progression of the degrading reaction under the UV (365 nm) (see Figure 5a) and visible (513 nm) (see Figure 5b) LED light irradiation, respectively, while the pH value was fixed at 6.5, taking into account the results of the zeta potential measurement and ensuring that no physical absorption of the MB dye molecules on the surface of the NPs. To have a background for comparison, the MB concentration in a solution sample with NPs at pH = 6.5 was measured in darkness (without irradiation) and no change was detected after a time period of 24 h, confirming that there is no physical adsorption of MB dyes on the NPs in accordance with the low NP concentration and the low electrostatic surface potential.

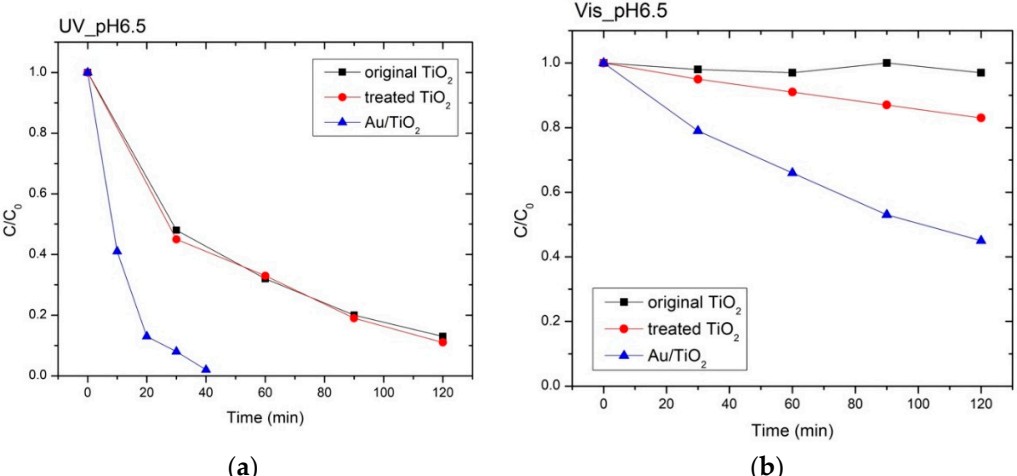

**Figure 5.** Comparison of the photocatalytic degradation of MB with different photocatalysts under (**a**) UV and (**b**) visible light irradiation at a pH value of 6.5.

As can be seen in Figure 5a, the samples with pristine $TiO_2$ NPs and the plasma-treated $TiO_2$ NPs show almost the same drastic decrease in the MB concentration, i.e., a high and almost identical photocatalytic degradation of MB under UV light. This implies that the plasma treatment with ammonia and hydroperoxy radicals only modified the surface, but not the crystal structure of the $TiO_2$ NPs.

The data in Figure 5b, on the other hand, shows distinctly different features for the three samples. There was no degradation of MB at all in the solution sample with $TiO_2$ NPs under visible light, whereas the plasma-treated $TiO_2$ NPs exhibited a mild but non-negligibly enhanced MB degradation performance under visible light. With the nitrogen atoms on their surface, the plasma-treated $TiO_2$ NPs can absorb visible light to a certain level, giving a mild photocatalytic effect under such irradiation. The sample with the $Au/TiO_2$ nanocomposite showed a decrease of $C/C_0$ down to about 40% at a reaction time of 120 min, denoting a substantially higher enhancement of the photocatalytic efficiency.

All the above presented results have revealed that the $Au/TiO_2$ nanocomposite exhibited the best performance regarding the photocatalytic degradation of MB here. Under UV light irradiation, the photocatalytic efficiency of the $Au/TiO_2$ nanocomposite was almost enhanced by 4.3 times when compared to that of the pristine and plasma-treated $TiO_2$ NPs, which is also a noteworthy enhancement of about 2.5 times of that obtained with the $TiO_2$-Au hybrid NPs in our previous work [36]. This enhanced photocatalytic effect is attributed to the charge separation effect of the $Au/TiO_2$ nanocomposite. A Schottky junction might be formed at the $TiO_2$/Au NPs interface, which creates an internal electric field (a space-charge region) inside the $TiO_2$ NPs [10]. This internal electric field acts to accelerate the UV-generated electrons and holes inside the $TiO_2$ NPs each in opposite directions, hence reducing the recombination rate of them due to movement. Thanks to that, the photocatalytic efficiency of the $Au/TiO_2$ nanocomposite was enhanced when compared to that of the original $TiO_2$ NPs. The better performance of the $Au/TiO_2$ nanocomposite in comparison to that of $TiO_2$–Au hybrid NPs can be attributed to the larger Au–$TiO_2$ interface in the $Au/TiO_2$ nanocomposite as already pointed out in the previous work [26].

The presented results also reveal that the $Au/TiO_2$ nanocomposites show the strongest photocatalytic activities under the visible light irradiation. The photocatalytic effect in the $Au/TiO_2$ nanocomposites under visible light is attributed to the LSPR effect of the AuNPs [12].

For the comparison of the catalytic efficiency of the plasma treated $TiO_2$ NPs and the $Au/TiO_2$ nanocomposite, the MB decomposition rates of these materials at three distinct different pH values are shown in Figure 6a–d. As expected from the results shown and discussed above, the MB decomposition rate of the treated $TiO_2$ NPs under the visible light shall be almost zero at pH = 3 and higher at pH = 10, compared to that at pH = 6.5. Due

to the highly positive zeta potential at pH = 3, positively charged MB molecules cannot approach the surfaces of the treated $TiO_2$ NPs. Because the lifetime of the $OH^\bullet$ radicals is short, the $OH^\bullet$ radicals generated via the hole oxidation of water at the surface of the NP photocatalysts under visible light can only decompose MB molecules which are close to their surfaces, and, thus, there will be no MB decomposition from the treated $TiO_2$ NPs under visible light (as can be seen in Figure 6a). Likewise, the MB decomposition rates of the $Au/TiO_2$ nanocomposites also increased with pH values as compared to the treated $TiO_2$ (see Figure 6b); however, at pH = 3, the $Au/TiO_2$ nanocomposite still shows a significant decomposition rate for MB under visible light. Since the zeta potential of the $Au/TiO_2$ nanocomposite was just slightly positive, a certain amount of MB molecules still approach its surface and may be decomposed by the $OH^\bullet$ radicals generated via hole oxidation.

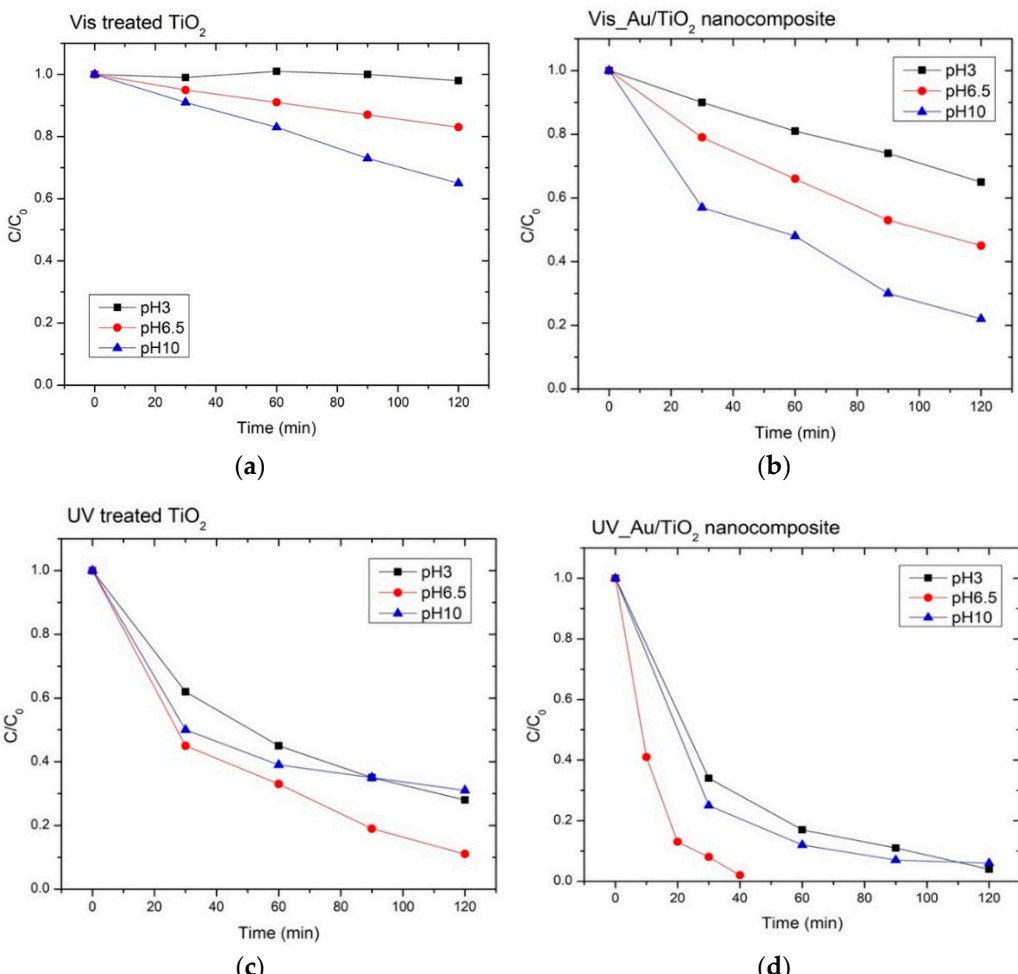

**Figure 6.** Photocatalytic degradation of MB under visible light irradiation with (**a**) treated $TiO_2$ and (**b**) $Au/TiO_2$ nanocomposite and UV light irradiation with (**c**) treated $TiO_2$ and a (**d**) $Au/TiO_2$ nanocomposite at different pH values.

Similar to our previous work, it is also evident in this study that the effect of the pH value on the MB decomposition rate under the UV light is distinctively different from that with visible light irradiation (see Figure 6c,d). As explained in the previous work [36], the electron reduction pathway plays the dominating role in MB decomposition at low pH values, whereas the hole oxidation pathway dominates at high pH values. Both pathways can operate at neutral pH values, giving rise to a greater MB decomposition rate for neutral pH scenarios.

## 4. Conclusions

Plasmonic photocatalysts of a Au/TiO$_2$ nanocomposite were synthesized by a microplasma-liquid interaction method here. The photocatalytic efficiency of the Au/TiO$_2$ nanocomposites under the UV and the visible light irradiation was compared with that of pristine TiO$_2$ NPs and the treated TiO$_2$ NPs by measurement of the photocatalytic degradation rates of MB dyes under different pH conditions. The synthesized Au/TiO$_2$ nanocomposite showed a strong enhancement of the photocatalytic activity under both UV and visible light irradiation. The treatment of the TiO$_2$ NPs by microplasma in the presence of NH$_3$ and H$_2$O$_2$ improved the attachment of the AuNPs on the Au/TiO$_2$ nanocomposite and, thus, enhanced the photocatalytic activity of the synthesized Au/TiO$_2$ nanocomposite. This improvement was attributed to the enlarged Au-TiO$_2$ interface. The plasma-treated TiO$_2$ NPs also showed photocatalytic activity under visible light irradiation.

**Author Contributions:** Investigation, N.T.T.T.; Project administration, L.H.M.; Resources, N.T.H.; Writing—original draft, P.H.M.; Writing—review & editing, D.H.T. All authors have read and agreed to the published version of the manuscript.

**Funding:** This work was financially supported by the International Center of Physics under the auspices of UNESCO under the Grant number ICP.2021.08 and by Vietnam Academy of Science and Technology under the Grant number QTBY01.04/20-21.

**Institutional Review Board Statement:** Not applicable.

**Informed Consent Statement:** Not applicable.

**Data Availability Statement:** Not applicable.

**Conflicts of Interest:** The authors declare no conflict of interest.

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
