# Peer review of "Improvement of the Photocatalytic Activity of Au/TiO2 Nanocomposites by Prior Treatment of TiO2 with Microplasma in an NH3 and H2O2 Solution"

_2571-8800, doi:10.3390/j5020019_

Round 1

Reviewer 1 Report

In the paper entitled "Improvement of photocatalytic activity of Au/TiO2 nanocomposite by prior treatment of TiO2 with microplasma in NH3 and H2O2 solution" the Authors presented the electrocatalytic synthesis of AuNPs-TiO2 nanosystems used for photodegradation of methylene blue. The obtained results are interesting and therefore should be published. However, a minor revision should be performed (see some issues below).

  1. Please perform the UV-Vis spectra of all obtained materials in transmittance mode in aqueous solutions.
  2. Why did the Authors use the green light for irradiation? Please explain.
  3. What about the photodegradation of the photocatalyst? Was it reusable? Please perform additional studies and calculate the TON of the photocatalyst.

Author Response

The authors would like to thank the reviewer's comments and answer the reviewer's questions as follows:

Q1.

Please perform the UV-Vis spectra of all obtained materials in transmittance mode in aqueous solutions.

Answer

It’s our mistake not to measure the transmittance of the samples before. Unfortunately the amount of sample is small we have not stored them to perform the measurement now.

Q2

Why did the Authors use the green light for irradiation? Please explain.

Answer

The green light is in the visible range and also in the strong absorption range of Au nanoparticles

Q3

What about the photodegradation of the photocatalyst? Was it reusable? Please perform additional studies and calculate the TON of the photocatalyst.

Answer

A very nice piece of extensive research. We'll look into that more deeply in the future.

Reviewer 2 Report

The manuscript describes the synthesis of Au/TiO2 composites in which the TiO2 support in a liquid was pretreated with an atmospheric micro-plasma. It was shown that the photocatalytic activity of the composite material pretreated in the microplasma is higher than that of the untreated material. The reason for the higher activity of the material pretreated in microplasma both upon irradiation with UV and visible light is assumed to be an increase in the gold-titanium oxide interface. The results are interesting and might be published after some revision.

It would be nice to present the DRS spectra of the three samples. Is there a plasmon band of Au nanoparticles visible?

Where were the Au nanoparticles formed during the microplasma treatment, mainly in the liquid phase? The size range of the Au nanoparticles on the TiO2 surface should be given.

The amount of photocatalyst produced by this method is very small. Is it possible to scale up the applied method? Is it possible to obtain the XRD powder pattern despite the small amount of catalyst obtained from the applied synthesis? They should be added to the manuscript.

The discussion refers only to the behavior of methylene blue. However, this is converted relatively quickly to intermediates which may interact weaker or stronger with the catalyst surface than methylene blue. The change in the shape of the UV/Vis spectrum with irradiation time suggests formation of such intermediates. This should be considered in the discussion.

A diagram showing the results of scavenger experiments for OH radicals, superoxide radicals, and holes should be added for Au/TiO2 (e.g., for pH 6.5) to hold experimental indication for the role of each active species.

How do the authors conclude that the presence of NH3 and H2O2 improves the attachment of Au nanoparticles. No results were shown with a material prepared with microplasma treatment in the absence of NH3 and H2O2.

Author Response

The authors would like to thank the reviewer's comments and answer the reviewer's questions as follows:

Q1

It would be nice to present the DRS spectra of the three samples. Is there a plasmon band of Au nanoparticles visible?

Answer

We have measure the DRS spectra of the samples and presented in our previous work

Appl. Sci. 2020, 10(10), 3345; https://doi.org/10.3390/app10103345

The plasmon band of Au nanoparticles is visible but shifted to longer wavelength, especially for plasma treated sample

Q2

Where were the Au nanoparticles formed during the microplasma treatment, mainly in the liquid phase? The size range of the Au nanoparticles on the TiO2 surface should be given.

Answer

As explained in our previous work Appl. Sci. 2020, 10(10), 3345; https://doi.org/10.3390/app10103345

For the case of nontreated sample the Au nanoparticles are formed in the liquid phase then grafted onto the TiO2 surfaces with the average diameter of about 30nm. However, for the treated one the Au nanoparticle mainly grow from the TiO2 surface and their average size is in the rage of about 8 nm.

Q3

The amount of photocatalyst produced by this method is very small. Is it possible to scale up the applied method? Is it possible to obtain the XRD powder pattern despite the small amount of catalyst obtained from the applied synthesis? They should be added to the manuscript.

Answer

XRD is worthy, however we cannot have access to this measurement. Please let’s save this investigation for further work

Q4

The discussion refers only to the behavior of methylene blue. However, this is converted relatively quickly to intermediates which may interact weaker or stronger with the catalyst surface than methylene blue. The change in the shape of the UV/Vis spectrum with irradiation time suggests formation of such intermediates. This should be considered in the discussion.

Answer

The referee’s comment is correct. However, in this study we only investigate on the degradation of methylene blue. Furthur work will be carried out with the investigation on COD (chemical oxygen demand)

Q5

A diagram showing the results of scavenger experiments for OH radicals, superoxide radicals, and holes should be added for Au/TiO2 (e.g., for pH 6.5) to hold experimental indication for the role of each active species.

Answer

Thank you very much for the comment. We have done that in our previous work

Applied Surface Science Volume 573, 30 January 2022, 151383

https://doi.org/10.1016/j.apsusc.2021.151383. This work we pay attention mainly to the effect of zeta potential.

Q5

How do the authors conclude that the presence of NH3 and H2O2 improves the attachment of Au nanoparticles. No results were shown with a material prepared with microplasma treatment in the absence of NH3 and H2O2.

Answer

We have conducted that study in our previous work Applied Surface Science Volume 573, 30 January 2022, 151383 https://doi.org/10.1016/j.apsusc.2021.151383. It’s shown that the plasma treated in the absence of NH3 and H2O2 has weaker photocatalyst ability in comparision to the sample in this present work